# Design and Calibration of Plane Mirror Setups for Mobile Robots with a 2D-Lidar

**DOI:** 10.3390/s22207830

**Published:** 2022-10-15

**Authors:** James E. Kibii, Andreas Dreher, Paul L. Wormser, Hartmut Gimpel

**Affiliations:** 1Department of Electrical Engineering and Information Technology, HTWG Konstanz, University of Applied Sciences, 78462 Konstanz, Germany; 2Department of Mechanical Engineering, HTWG Konstanz, University of Applied Sciences, 78462 Konstanz, Germany; 3Institute for Optical Systems, HTWG Konstanz, University of Applied Sciences, Alfred-Wachtel-Straße 8, 78462 Konstanz, Germany

**Keywords:** Lidar, environmental perception, mirror, calibration procedure, mobile robot, ground detection

## Abstract

Lidar sensors are widely used for environmental perception on autonomous robot vehicles (ARV). The field of view (FOV) of Lidar sensors can be reshaped by positioning plane mirrors in their vicinity. Mirror setups can especially improve the FOV for ground detection of ARVs with 2D-Lidar sensors. This paper presents an overview of several geometric designs and their strengths for certain vehicle types. Additionally, a new and easy-to-implement calibration procedure for setups of 2D-Lidar sensors with mirrors is presented to determine precise mirror orientations and positions, using a single flat calibration object with a pre-aligned simple fiducial marker. Measurement data from a prototype vehicle with a 2D-Lidar with a 2 m range using this new calibration procedure are presented. We show that the calibrated mirror orientations are accurate to less than 0.6° in this short range, which is a significant improvement over the orientation angles taken directly from the CAD. The accuracy of the point cloud data improved, and no significant decrease in distance noise was introduced. We deduced general guidelines for successful calibration setups using our method. In conclusion, a 2D-Lidar sensor and two plane mirrors calibrated with this method are a cost-effective and accurate way for robot engineers to improve the environmental perception of ARVs.

## 1. Introduction

### 1.1. Lidar Sensors on Vehicles

Light detection and ranging (Lidar) sensors generally measure the distance to an object by emitting laser light towards the object and then observing the time of flight until some of the backscattered light returns to the sensor. This Lidar measurement principle can be turned into different kinds of optical sensors: either a laser range sensor statically directs its light into a single direction and, therefore, measures a single distance value for any object in this direction or a 2D laser scanner, also called single-layer laser scanner or 2D-Lidar, additionally uses one mechanical axis of rotation to spin the emitted laser light, up to a full 360° rotation. This kind of 2D-Lidar has been commercially available for several decades. During the last few years, 3D-Lidar sensors have also become available. These can roughly be separated into two groups: The group of spinning 3D-Lidar sensors, also called multiple-layer laser scanners, still uses a macroscopic mechanical rotation axis to spin a multitude of emitted laser light beams, generally using a motor. They can have a field of view (FOV) of up to 360° along their scanning direction and typically around 40° in the perpendicular direction. The other group consists of 3D flash Lidar sensors, some of which are also called 3D solid-state Lidar sensors. These sensors avoid the macroscopic mechanical spinning movement by a multitude of different alternatives. A common property of these 3D flash Lidar sensors is their camera-like FOV, typically 40° in both axes. A recent overview of 3D-Lidar sensors for automotive applications can be found in [1].

Lidar sensors are commonly used on different kinds of vehicles for environmental perception [2] because of their advantages over Radar and camera sensors: Lidar sensors usually have a higher spatial resolution than Radar sensors, because of the shorter wavelength of light compared to radio-frequency Radar waves. Lidar sensors also incorporate a well-adapted light source and, thus, can still operate reasonably well under challenging conditions for optical sensors, such as darkness, bright sunshine, or moderate fog and rain. Compared to triangulating optical sensors, for example stereo cameras, Lidar sensors have the advantage that the distance resolution stays nearly constant even at the maximum object distance.

Even though 3D-Lidar sensors have attracted more attention, especially for their use in autonomous passenger vehicles, there are currently many application areas where 2D spinning Lidars are still predominantly used. One example of this are autonomous guided vehicles (AGVs), also called autonomous mobile robots (AMRs) [3]. The use of 2D-Lidars on AGVs is still reasonable because they already supply extremely useful data and their price is currently still much lower than a 3D-Lidar. Additionally, the smaller amount of point cloud data generated by these sensors makes data processing less challenging; it can be achieved with less expensive hardware and, therefore, further reduces costs.

The data from 2D-Lidars on AGVs are used for several purposes: firstly, for localization of the vehicle inside a known room; secondly, for navigation purposes, so the vehicle can plan and find a path to where it is ordered to move; thirdly, Lidar data are often used for mapping. This means that during the operation of the AGV, data are stored and used to continuously generate or update a map of the room in which the AGV moves. This is performed with various simultaneous localization and mapping (SLAM) algorithms. A fourth purpose is collision avoidance: the Lidar data help the AGV not to collide with fixed or moving objects.

A problem that arises when using 2D-Lidar sensors on AGVs is the lack of ground detection. To achieve long-distance sight, as well as to acquire and perceive map data on a fixed height above the ground, the scanning plane of a 2D-Lidar sensor is generally oriented parallel to the ground on which the AGV drives. Therefore, the ground itself cannot be detected by the 2D-Lidar sensor. This becomes especially problematic when the ground is missing, for example when the AGV approaches a staircase to a lower floor. In this case, the AGV could fall down those stairs and damage itself, as well as its environment. Another problem is obstacles on the floor that are not high enough to be detected by the scanning laser beam, but are still big enough to seriously stop or damage the AGV when it simply tries to drive over the obstacle.

This problem of missing ground detection can be solved by adding a separate sensor to detect the ground surface. This can be an ultrasonic distance sensor or an additional optical distance sensor such as a short-range laser range sensor. However, these additional sensors add further cost to the AGV hardware. In addition, the data quality and usefulness of the above-mentioned additional ground detection sensors is much weaker than the 2D-Lidar data. However, the ultimate solution of adding a second 2D-Lidar for ground detection or using a 3D Lidar both add significant cost.

Therefore, reshaping the FOV of a 2D-Lidar sensor to simultaneously achieve a wide range and ground detection is an important issue for improving 2D-Lidars on vehicles.

### 1.2. Previous Work on Lidar Sensors with Reshaped FOV

Several research groups have already added plane mirrors next to a Lidar sensor to reshape the FOV of the sensor. By this approach, a single Lidar sensor can additionally be used for ground detection, avoiding the need for a second sensor. This seems especially reasonable when a 2D-Lidar sensor with a 360° FOV used on an AGV is mounted at a position, e.g., at the very front of the AGV, where some of its FOV would otherwise be blocked by the presence of the vehicle. This use of mirrors to reshape the FOV of an optical sensor is similar to what has been done before with cameras looking simultaneously into several directions [4,5] or generating an omnidirectional FOV with catadioptric optics in front of a camera objective lens [6,7].

A conference paper from the year 2001 [8] is probably the first publication on the topic of Lidar sensors with additional plane mirrors. Two plane mirrors are attached to a 2D-Lidar sensor mounted at the front of an automobile. The resulting Lidar data were presented, as well as possible future applications sketched. A method for the calibration of the mirror positions was not mentioned in this publication.

This idea was not described in more detail until in 2017, when a research paper by Abiko et al. [9] used plane mirrors attached to a 2D spinning Lidar mounted on an unmanned aerial vehicle to help a quadcopter fly at a constant distance below a road bridge. This setup is very similar to the one presented here. However, Abiko et al. did not deal with the task of geometrical calibration of the mirrors either.

A later research paper [10] by Matsubara et al. used fixed plane mirrors next to a spinning 3D Lidar sensor on an autonomous vehicle to extend the sensor FOV to the ground. They also attached plane mirrors onto a manipulator arm on a vehicle [11] to improve its stair climbing ability. The work of this group differs from ours because they used a 3D spinning Lidar as opposed to a 2D spinning Lidar in our paper. In addition, they also did not deal with the calibration of the mirrors.

In a paper by Aalerud et al. [12], the FOV of a 3D spinning Lidar was reshaped by means of eight plane mirrors. The goal was to generate a similar FOV to that of a 3D flash Lidar, but using a 3D spinning Lidar, and to improve the spatial resolution. This group also presented mechanical design considerations for the placement of the mirrors and performance figures of the resulting prototype, and they stressed the need for a calibration method [12] (p. 22) to increase the precision of the measurement.

We know of only two publications that addressed the calibration process of the involved mirror positions and orientations: In a patent [13] by Waymo LLC, a manufacturing company of 3D Lidars, the need for such a calibration was mentioned briefly. In their context, the mirrors are far (meters) away from the Lidar sensor and the calibration is meant to track the position variations of the mirrors during vibrations that occur on a driving vehicle. Presumably, this ongoing calibration uses fiducial markers attached to the mirrors.

Recently, a paper [14] by Pełka et al. described a calibration procedure for a 3D flash Lidar where the FOV is reshaped by six plane mirrors into a shape that is similar to that of a 3D spinning Lidar. Their presented calibration method does not use any fiducials on the mirrors and relies on “pairs of the nearest neighborhood points that were reflected by a different mirror” [14] (p. 9). This approach cannot succeed with data from 2D-Lidar sensors, as in our proposed configuration, because there can be no more than one of these point pairs at the position where two scan lines of the 2D-Lidar cross each other. From our experience, one single point pair is insufficient for calibration without additional information such as fiducials or similar aids.

Therefore, to our knowledge, so far, no calibration procedure has been presented to acquire the position and orientation of plane mirrors attached to a 2D-Lidar sensor.

### 1.3. Aim

The aim of this work was to develop an easy-to-perform calibration procedure for a setup of a 2D-Lidar sensor with two attached plane mirrors. Accurate calibration is necessary because, usually, the position and orientation values that can be taken from the computer-aided design (CAD) software used to design the mechanical parts that hold the mirrors are not precise enough to measure accurate point cloud data. Production tolerances in the mechanical parts themselves, as well as individual variations during the fixation process of the optical mirror part, e.g., by gluing, are causes for deviations in orientation and position from the ideal CAD values.

We aimed to develop a calibration method that uses a simple calibration object and that can be used cost effectively in industrial production processes, as well as by researchers or amateurs when building a mobile robot with a 2D-Lidar and a two-mirror setup. This goal is complicated by the fact that the mechanical system to calibrate has many degrees of freedom, but point cloud data acquired with a 2D-Lidar on calibration objects of a simple geometry do not provide many features to be used for the calibration algorithm.

Additionally, we explored different design possibilities for the geometric arrangement of two plane mirrors around a 2D-Lidar sensor in order to add ground-detecting capabilities to the vehicle. We give several examples for geometric setups that are suitable for different purposes and boundary conditions that may be encountered on an AGV.

## 2. Materials and Methods

### 2.1. Prototype Vehicle

The presented measurements were conducted with a prototype vehicle that consisted of a “PiCar-S” [15] by SunFounder, not using any of the included sensors, and a “Raspberry Pi 4B” computer [16] running Linux (Raspberry Pi OS) and Python 3 [17]. A 2D scanning Lidar “TIM310-S01” [18] by Sick AG was connected via USB to the vehicle’s computer. This Lidar has an angular FOV of 270° with a resolution of 1°, a maximum measurement distance of 2 m on targets with 10% remission, a scanning frequency of 15 Hz, and an IR (infrared) laser wavelength of 850 nm. The computer and the Lidar sensor were attached on top of the vehicle, while two rechargeable lithium polymer batteries and a step-up voltage converter were attached at the bottom of the vehicle for power supply.

Two plane rectangular first surface mirrors [19] by Edmund Optics measuring 100 × 75 mm with a 3 mm tick float glass substrate and a “protected silver” reflection coating were attached at either side of the Lidar sensor. The nominal reflectivity of these mirrors at an 850 nm wavelength was specified as >98%. In the prototype vehicle, only a fraction of the existing mirror surface is actually used for reflecting the Lidar’s laser beam; the mirror dimensions are only this large because the mirrors were bought as off-the-shelf parts. A metal reflection coating was chosen so that mirror reflectivity does not strongly depend on the angle of incidence of the light for a large range of incidence angles. For holding these mirrors at the intended positions, mechanical parts were designed in the CAD software “Creo Parametric 8” [20] and produced as fused deposition modeling parts on a 3D printer out of polylactic acid (PLA) polymer material. The mirrors were attached to the mechanical parts by “Fixogum” rubber cement.

A picture of the described prototype vehicle is shown in Figure 1. This vehicle also has a simple driving algorithm implemented in Python to perform manual and automatic driving on the base of the Lidar data, which is not the topic of this article. A video of the driving vehicle can be seen in the Appendix A to this article.

### 2.2. Geometric Position and Orientation of Mirrors

Extensive CAD simulations were performed regarding the different possibilities to position and orient the two plane mirrors next to the Lidar sensor. These simulations were performed with the parametric CAD software “Creo Parametric” [20]. In Figure 2a, a top view projection of the geometry used is sketched. The green circles drawn around the mirror edges have a diameter of 27 mm; this corresponds to the light receiving aperture of the Lidar sensor used here. Whenever the red line of an observation direction touches a green circle, this corresponds to a power loss of the received Lidar light of 50%, because only half of the receiving aperture of the Lidar sensor can then receive the light reflected at this mirror edge. For this reason, the CAD model was parametrized such that these observation direction lines always keep a distance of 13.5 mm from the mirror edges, as well as from the housing of the Lidar sensor.

The resulting reshaped Lidar FOV can be seen in Figure 2b. Observation angles that are not reshaped by any mirror make up the front view of the vehicle, whereas each of the two mirrors reflects a piece of the original 2D FOV towards the ground. As a result, two line-shaped pieces of the FOV observe the ground. These two line-shaped mirror view FOVs should cross each other somewhere on the ground to ensure a gapless observation of the ground in front of the vehicle.

The Cartesian coordinate system used has its origin in the central rotation axis of the Lidar sensor at the height of the center of the receiving aperture of the Lidar. The y-axis points into the direction of movement of the vehicle, the x-axis towards the right mirror (seen from the top), and the z-axis towards the top of the vehicle.

The definitions used to describe the geometric setup of the two mirrors, as well as the resulting key performance figures of the geometries are visualized in Figure 3. All CADs were mirror-symmetrical around the yz-plane. The angle α describes the FOV of the scanning Lidar that is not disturbed by the mirrors; this part of the FOV generally points to the front of the vehicle. The angle β is the tilt angle of both mirrors around the global z-axis, and γ is their inclination angle around the global x-axis. The length L_1_ is the total mechanical width of the mirror setup; L_2_ is the minimum distance between the front of the vehicle and the detectable part of the ground; L_3_ is the maximum distance between vehicle and the detectable ground. Finally, L_4_ is the distance from the center of the Lidar to the front of the vehicle measured along the y-axis.

The following additional geometric quantities were not changed during the CAD simulations, because they were determined by the prototype vehicle and the properties of the Lidar sensor: the width of the vehicle was 135 mm; the height of the Lidar (Cartesian origin) above the ground was 160 mm; the diameter of the nearly cylindrical Lidar housing in the xy-plane was 38 mm.

For the experiments to calibrate the geometric position of the mirrors in Section 3.3, one fixed geometry configuration was used. The corresponding CAD values of all geometric parameters during those experiments are listed in Table 1. This geometric setup was chosen as a good compromise between most of the other explored geometric setups.

For the data in Section 3.1, the CAD model of the prototype vehicle was used to verify that the angle of incidence of the laser beam on the two mirrors stayed below 60° for all Lidar rotation positions that contributed to the Lidar measurement data. Therefore, no significant power loss of the laser beam during reflection on the metallic mirror coatings was expected.

During the CAD, no special care was taken to avoid a direct back reflection of the Lidar sensor’s emitted laser light into the receiving aperture of the Lidar. This problem was reported in [10] (p. 4), but was not observed with the present setup.

### 2.3. Point Cloud Transformation Algorithm

The 2D-Lidar sensor returns a distance value for each rotation angle inside its 270° FOV. Depending on the angular position of the Lidar rotor, this distance value needs to be transformed differently into a triplet of x-, y-, and z-coordinates, the so-called point cloud data.

For Lidar viewing angles around the forward direction of the vehicle, the laser beam was not deflected by any mirror, so the returned distance value can be directly taken as the object distance for this viewing angle. These data values only undergo a simple coordinate transformation from the raw angle–distance value pair in polar coordinates into a 3D Cartesian coordinate vector. In this case, the resulting point cloud data remain on the x-y plane per definition.

For Lidar viewing angles where the laser beam and the returned light were reflected by one of the mirrors, a slightly more complex transformation needs to be calculated. The resulting point cloud data after the transformation depend on the distance and angular orientation of the mirror that reflects the light. The algorithm of this point cloud transformation as presented in Formulas (1) to (4) was taken from [12] (p. 5), but the formula (2) for r was corrected. For a visualization of the underlying 3D geometry, the figure in the given reference can be used. First, the Lidar viewing angle and measured distance value are transformed into a Cartesian coordinate vector d of the apparent object point. The geometric plane of the mirror surface is described with a support vector w somewhere on this plane and a normal vector n. The vector to the reflection point I on the mirror is calculated as
(1)$$I=\frac{n\cdot w}{n\cdot d}\;d$$The unit vector r of the reflected light beam is
(2)$$r=\frac{d}{\left|d\right|}-2\left(\frac{d}{\left|d\right|}\cdot \frac{n}{\left|n\right|}\right)\;\frac{n}{\left|n\right|}$$
and the vector v from this reflection point to the transformed data coordinate point is
(3)$$v=\left(\left|d\right|-\left|I\right|\right)\;r$$This finally makes the vector d′ the transformed data point
(4)$$d^{\prime }=I+v$$

There are some Lidar viewing angles where the laser beam is emitted in a direction that gets close to an edge of either of the mirrors. The corresponding viewing angles of the Lidar were precalculated, and Lidar distance data from those directions were not used for further data analysis. The interval of these omitted viewing angles was calculated such that at least 50% of the receiving optical aperture of the Lidar always had a clear beam path either on one of the mirrors or missing both of the mirrors.

### 2.4. Calibration Setup and Procedure

Our calibration procedure consisted of the following easy steps, which are summarized in Figure 4 and detailed below:
Determine the distance between each mirror and the Lidar along a fixed direction;Position the flat calibration object in front of the Lidar;Acquire and save Lidar distance data on this flat calibration object;Calculate the position and orientation of both mirrors that fit the data best.

In the first step, the mirrors were covered with white flat paper, e.g., using adhesive film. Lidar distance data were acquired for several hundred rotation periods of the spinning Lidar. For each mirror, the arithmetic mean value of the measured distances along one fixed direction towards this mirror was taken. In our geometry, both distances dr and dl to the two mirrors were determined in the x-direction. The paper was then removed from the mirrors after this measurement.

For the second step, a flat calibration object was used, which consisted of a flat sheet of material that had a homogenous, light, and diffuse surface. Our experiments were performed with a 4 mm-thick sheet of polymer material with size 700 × 700 mm. which was entirely coated with white paint on its front side; see Figure 5. Additionally, a small square piece of retro-reflection foil was attached roughly to the center of this calibration object. The size of this reflector foil was chosen such that it corresponded to the angular resolution of the Lidar sensor in the intended calibration distance. For our experiment, a 10 mm square piece of white-color “Diamond Grade” reflective sheeting [21] was used, but any other retroreflecting foil should work equally well. The position of the retroreflector can later be recognized easily in the acquired Lidar data because the sensor used in this experiment additionally returns the brightness value for every returned distance value. This brightness value is much higher on the reflector material than on the surrounding white surface. In the case of using a different Lidar sensor, which does not return brightness values, this reflector foil can be omitted, and a hole of a similar size can be made in the flat calibration object instead.

The calibration target needs to be positioned with the retroreflection foil at the crossing point where the laser beam passes twice, once reflected by one of the mirrors and also reflected by the second mirror. To achieve this, the Lidar was turned on and the light lines of the Lidar sensor’s IR laser on the calibration target were observed with an IR observation tool while positioning the calibration target. In our experiments, a hand-held IR viewer was used for this purpose, but a smartphone camera with an imperfect IR blocking filter can alternatively be used to visualize this in a dark room.

As the third step, Lidar data with this well-positioned calibration object were acquired and saved to a data file. Usually, several hundred rotation periods of the spinning Lidar are used; this typically corresponds to a few minutes of data acquisition time.

For step four, we transferred this acquired Lidar data from the vehicle’s computer to a different computer for convenience, but the following data processing steps could in general also be performed on the Raspberry Pi computer on the vehicle itself. The acquired data were used in the algorithm detailed below to accurately determine the exact orientation angles of both mirrors in the setup. This calibration algorithm was implemented in Python [17] and mainly relies on an implementation of the downhill simplex algorithm, which is available in Python’s SciPy library [22]. This algorithm minimizes the value of a scalar function of multiple arguments. A problem-specific scalar error function was defined as detailed below. The free variables in this error function were the two independent orientation angles of both mirrors, as well as the distance and orientation of the flat calibration target. These free variables were parametrized with support (s) and normal (n) vectors as follows:
Normal vector of the right mirror surface: nr = (nr_x_, nr_y_, −1);Normal vector of the left mirror surface: nl = (nl_x_, nl_y_, −1);Normal vector of the flat calibration target: nt = (nt_x_, −1, nt_z_);Support vector of the flat calibration target: st = (st_x_, st_y_, st_z_); this vector was also taken to be the position vector of the retroreflector foil on the target.

Setting one coordinate value of the above normal vectors to −1 does not restrict the generality of the chosen parametrization because none of these planes are oriented perfectly perpendicular to any of the coordinate axes. The minus sign was chosen to describe the expected orientation in the given coordinate system.

Additionally, the following fixed parameters were used as determined in Step 1:Support vector of right mirror surface: sr = (dr, 0, 0);Support vector of left mirror surface: sl = (−dl, 0, 0).

The problem-specific error function *err* to be minimized during the optimization of the above mirror orientation parameters is defined as the sum of three positive contributions:(5)$$err=err_{RMS}+err_{r}+err_{l}$$The first contribution *err**_RMS_* to this error function in Formula (5) is defined as the root-mean-squared (RMS) distance of the transformed point cloud (see Section 2.3) to the calibration target plane. For this RMS value, the data from several turns of the spinning Lidar sensor were used. Only data points that were known either to hit the calibration target directly or to be reflected properly by one of the two mirrors were considered. The transformed point cloud data still depends on the two mirror orientations. The distance of a point p to a plane $E_{s,n}$ defined by a support vector s and a normal vector n is calculated according to Euclidean geometry [23] as
(6)$$dist\left(p,E_{s,n}\right)=\left|\left(p-s\right)\cdot \frac{n}{\left|n\right|}\right|$$

Before the other two contributions *err_r_* and *err_l_* to the error function in Formula (5) can be evaluated, two angle parameters need to be calculated first. The parameter α_refl,r_ is the angular position of the Lidar’s rotor when viewing the retroreflector foil on the target via a reflection on the right mirror. The corresponding parameter α_refl,l_ is the angular position of the Lidar when viewing the reflector foil via a reflection on the left mirror. Both angular positions can be calculated from the saved calibration dataset by searching for data points that have an extraordinarily high brightness value returned by the Lidar sensor. If the previous positioning (Step 2 of the calibration procedure) succeeded, then the dataset included several of those points for reflections over both mirrors. Only data points observed under α_refl,r_ were considered for the calculation of the error function contribution *err_r_* and, accordingly, for *err_l_*. Both of these error function contributions were then calculated as the RMS distance value of all the considered data points with a high brightness value to the position of the support vector st of the calibration target.

Since both types of error function contributions were measured in units of length, the calculations were performed in millimeters during our experiments, and since they also were of a similar size, no scaling prefactors were used in the definition of the error function in Formula (5). For other Lidar sensors or different angular resolutions, it could be useful to add a scalar prefactor in front of the *err**_RMS_* term in Formula (5) in order to scale both types of contributions to a similar magnitude.

Eventually, with this problem-specific error function defined, the minimization algorithm was used to determine the optimal normal vectors of both mirror surfaces that gave the smallest possible value for this error function. Initial values for all optimization parameters were taken to be CAD values or even rough measurement values acquired on the prototype vehicle with a measuring tape. We allowed the optimization algorithm a maximum number of 100,000 iterations, and it normally took a few minutes on a common laptop computer for it to finish.

The described marking of the intersection point of the two reflected Lidar FOV sections with a piece of bright retroreflector foil proved to be a very useful additional constraint during the parameter optimization. Measuring the mirror distance to the Lidar beforehand with an independent measurement (Step 1) also proved to be a useful and necessary addition to the procedure, as it reduced the number of free parameters to be optimized. Leaving out either of these two described extra steps resulted in a highly unstable calibration and often also in the wrong calibration results.

## 3. Results

### 3.1. Prototype Vehicle Performance

Three relevant performance properties of our prototype vehicle were evaluated experimentally as follows: First, we checked how the reflection of the Lidar’s laser light at the mirrors affected the acquired distance data. For this purpose, two fixed observation angles of the spinning Lidar were chosen. In one of these directions, the Lidar’s light was reflected off the plane mirror and directed towards a flat object of homogenous brightness. In the second chosen direction, the Lidar sensor measured data from a direction where the same flat object was observed at mostly the same distance. For both directions, 1000 distance values were acquired. The measurement results were:
Direct view (no mirror): mean distance 403.2 mm, standard deviation 1.9 mm;View reflected by mirror: mean distance 400.4 mm, standard deviation 2.1 mm.

The resulting standard deviation of the measured distance without any mirror is typical for the 2D-Lidar sensor used. When reflecting the light with a mirror, the standard deviation increased by 10%. This small increase did not significantly deteriorate the Lidar’s distance measurement performance. It was probably due to the loss of light intensity during the reflection on the mirror. The mirror datasheet states a reflectivity of R > 98% for an angle of incidence of 45°; it is plausible that for the used angle of incidence (<60°) in the prototype setup, this power loss is higher. Additionally, the optical power loss is experienced twice during Lidar operation, once by the emitted laser light beam and once again by the returning light.

To compare this small observed increase in distance noise by a mirror reflection to other environmental effects on the distance noise, a third measurement was performed on a different target object with much lower surface roughness:
Direct view (no mirror), less object roughness: mean dist. 404.5 mm, st. dev. 4.5 mm.

This confirmed the conclusion that the distance noise increase that was due to the mirror reflection was much smaller than distance noise increases by other environmental influences, such as this smoother surface of the observed object. Therefore, we concluded that the mirrors used in the prototype setup were suitable for the chosen purpose, because they only increased the distance noise of the measured distance values by 10%.

A second check performed was to verify the actual size and position of the reshaped FOV on the ground in front of the prototype vehicle. The geometric parameters used in this vehicle, as well as the resulting theoretical FOV size on the ground from the CAD calculations are listed in Table 1. A hand-held IR viewer and a measuring tape were used to determine experimentally the positions where the Lidar light can be seen on the ground in front of the vehicle. The results are presented in Table 2.

The results in Table 2 underline the need for a proper calibration of the mirror positions and orientations on the vehicle. The measurement results were similar to what the CAD calculation predicted, but they still differed by significant amounts up to 50%. Additionally, it can be seen that the left and the right mirror mounted onto the vehicle differed in their orientation, because the left and right measured FOV positions were not identical.

In a third experimental check, obstacles were placed on a flat and horizontal ground in front of the prototype vehicle, and the recorded Lidar data were visualized. Here, the intention was to find out for what size of obstacles the prototype setup can be useful. Little polymer cubes were used as obstacles, with sizes between 5 mm and 30 mm. A picture of the measurement setup can be seen in Figure 6a and the resulting Lidar distance data from a single Lidar rotation period in Figure 6b.

In all the resulting Lidar raw distance data of Figure 6b, it can be seen that the distance value decreased for increasing Lidar rotation angles. This is only an artefact of the geometric configuration where the same flat ground in front of the vehicle was further away when observed by mirror reflections under different Lidar rotation angles. The slight deviation of the distance values from a straight line was due to the inherent distance noise of the Lidar data. The dip in these decreasing curves was caused by the obstacle cube. It can be seen that cubes of sizes between 30 mm and 10 mm caused distance variations, which can readily be distinguished from the distance noise. Thus, obstacles of these sizes could be detected by the presented setup in the Lidar data taken during just one single Lidar rotation period. The dip in the data for a cube size of 5 mm was very slight and could therefore not be reliably distinguished from random distance noise. Therefore, cubes of size 5 mm could not be reliably detected with the present setup during a single rotation period of the Lidar. (This experimental conclusion assumed that a straight line is fit to the data points in Figure 6b; some acceptance corridor around this line is defined such that the small distance deviations from this line due to noise were not considered, but the visible larger distance deviations from this line due to the obstacles can be recognized.) Nevertheless, the distance data could be averaged over several turns of the Lidar rotation, thus decreasing the distance noise of the averaged distance values. This could probably even make the detection of 5 mm-sized objects possible with the present setup, but would result in a longer reaction time, because the necessary data acquisition for this averaging obstacle detection algorithm would take the time of several Lidar turns. For the presented prototype vehicle, this averaging strategy was not pursued, because the vehicle was capable of driving over obstacles of this size without noticeable damage. In the Appendix A to this article, some videos show how the vehicle drives and avoids obstacles.

### 3.2. Geometric Design Results

To explore the wide range of geometric possibilities when using a 2D-Lidar and two mirrors for ground detection, several different geometry configurations were simulated in the CAD software. All simulated configurations used the same vehicle and the same 2D-Lidar sensor. However, the mounting position and orientation of both mirrors was changed, while still keeping the Lidar sensor mounted at a position on the central yz-plane. For most geometry configurations, also the size of the two plane mirrors was kept unchanged, with the exception of the geometry aiming for a minimum vehicle width. The mounting position of the Lidar sensor on top of the vehicle was also changed. A prerequisite was that each FOV covered the ground at least over the width of the vehicle without blind spots.

The five different design goals were:
Each mirror covers the near ground on the opposite side;Each mirror covers the near ground on the same side;Angular FOV in the forward direction (parameter α) as large as possible;Total width of the vehicle (parameter L_1_) as small as possible;Width of the mirrors as small as possible.

For Goals 1, 2, and 4, the position of the Lidar was fixed centrally on the vehicle, while for Goals 3 and 5, it could be moved forward and backward.

In each of the above design goals, there were still many other geometric degrees of freedom available that led to a multitude of possible geometric setups. Therefore, for each of these design goals, one typical geometric solution was chosen that seemed to achieve the aim without going to unrealistic extremes in any other geometric parameter. The characteristic geometry parameters of each of our chosen CAD solutions are listed in Table 3. A top view of each CAD solution is shown in Figure 7.

The CAD variant for Design Goal 1 (Figure 7a) was identical to the prototype vehicle used here. This geometry was used throughout the measurements for this article because it is a good compromise between all the other presented geometries.

In general, there are two different strategies for reshaping the FOV of a 2D-Lidar sensor in order to achieve ground detection. This can be seen when comparing Design 1 to Design 2. The first possibility, as in Design 1 (Figure 7a), is to make the laser beam that hits the mirrors at the outward end reflect mostly into the driving direction of the vehicle, i.e., along the y-axis. This results in the right mirror illuminating the near ground at the left side in front of the vehicle and vice versa. The second possibility, as in Design 2 (Figure 7b), is to make the laser beam that hits the mirrors at the inwards end reflect mostly into the driving direction of the vehicle. This results in the right mirror illuminating the near ground at the right side in front of the vehicle and vice versa. When comparing the presented two geometry results, it can be seen that Geometry 1 generated a FOV on the ground that was much nearer to the front of the vehicle (L_2_ is smaller) and roughly had the width of the vehicle along its full extent in front of the vehicle. In contrast, Geometry 2 created an FOV on the ground with the nearest position further away from the front of the vehicle, but with a width that increased with the distance from the front of the vehicle.

The resulting geometry (Figure 7c) for Design Goal 3 proved that it was also possible to keep a large unaffected viewing angle α towards the driving direction of the vehicle, in the displayed example of 180°. For mounting positions of the 2D-Lidar at the very front of a vehicle, as is common for AGVs, this is the maximum useful viewing angle, because towards the back, the vehicle itself normally blocks the Lidar’s view. Therefore, the additional FOV on the ground can in this case be achieved without losing any relevant FOV towards the front.

Design Goal 4 was to minimize the total width L_1_ of the mirror setup. This can be necessary when space for adding plane mirrors next to a Lidar is limited. The corresponding result (Figure 7d) had a total width L_1_ of around 160 mm. Compared to the width of the current and typical 2D-Lidar sensor housing of around 70 mm, we considered this size increase by a factor of 2 acceptable.

In all previous designs presented, the size of the mirrors was kept constant. However, as can be seen in Figure 7, the 100 mm length of the reflecting surface of the mirrors was not fully used in most cases. This was the motivation for decreasing the size of the mirror parts in Design Goal 5. The result is presented in Figure 7e, where the length of the used mirror surfaces was reduced to 43 mm. This significantly reduced the weight of the resulting glass mirror setup. At the same time, it was possible to achieve a very narrow setup with L_1_ of only 168 mm in the presented example.

### 3.3. Calibration Results and Verification

The calibration procedure presented in Section 2.4 was experimentally tested for several different distances and orientations of the simple flat calibration object. The aim of these experiments was to check if the numeric optimization converged to a realistic result of the mirror orientations. For realistic calibration outputs, we additionally verified that the resulting geometry parameters also yielded realistic results for independent point cloud data, not already used during the calibration.

The flat calibration target with retro-reflector foil attached was positioned in the following different distances and orientations in front of the prototype vehicle. The given values were only rough estimates; more precise values were acquired as the output of the calibration procedure, but were not relevant to the following conclusions:
Setup A: target tilted ~45° around x-axis, ~0° around z-axis, distance ~300 mm;Setup B: target tilted ~30° around x-axis, ~0° around z-axis, distance ~300 mm;Setup C: target tilted ~45° around x-axis, ~20° around z-axis, distance ~300 mm;Setup D/E/F: same orientation as Setup A/B/C, larger distance ~400 mm;Setup G/H/I: same orientation as Setup A/B/C, larger distance ~600 mm.

For each of the above calibrations setups, the following quality criteria, chosen in analogy to the criteria used in [12] (p. 11), were evaluated:

The RMS absolute distance between the acquired points and the location of the calibration target plane; see Equation (6). All Lidar rotation angles that produced data on the calibration target, as well as the data from several turns of the spinning Lidar were considered for the calculation of this criterion.The arithmetic mean intensity value of the acquired points on the calibration target, in received signal strength indicator (RSSI) values as given by the Lidar sensor.The number of successfully acquired data points, given as a percentage of the number of measured data points. (The Lidar sensor can mark return data as invalid.)The standard deviation of the measured object distances into one fixed direction directly in front of the vehicle.

The percentage of acquired data points resulted in 100% for all our measurements on the white calibration target; therefore, this value is not stated individually in the following results.

The measured mean (± standard deviation) values for the mirror distances were:
Distance right mirror: dr = (83.3 ± 1.9) mm;Distance left mirror: dl = (82.5 ± 1.9) mm.

For each of these calibrations setups the results are given in Table 4. Additionally, for each setup, the calibrated mirror positions from the first line were used to calculate the quality criteria for the dataset of the second line. This was meant to be a first verification check. However, the resulting quality criteria values always were the same as in the first line. Therefore, this did not yield meaningful results, and these are not shown in Table 4.

For the ease of comprehension, the resulting values for the left and right mirror normal vectors (nr, nl) were transformed into an angle β, describing the tilt of the mirror around the global z-axis, and an angle δ, describing the tilt of the mirrors around the global x-axis. This β angle is defined identically to the β angle in the above CAD definitions and, thus, can be compared directly. The resulting (global) δ angle is defined differently as the (local) γ angle used above in the CAD definitions. Both angles, β and δ, are presented in Table 4. For the ease of comparison, the theoretical CAD values of the mirror angles were transformed to the global (δ) angle and displayed there as well.

The calibration setups A, B, D, E, and G produced δ angles that were more than a few degrees away from the respective CAD values. Those calibration setups were therefore considered unreliable, because the real prototype setup is known to have mirror angles that differ no more than a few degrees from their CAD values. The calibration setup F resulted in double the RMS distance value of the total point cloud compared to the distance standard deviation of only the point in the front direction. This was also taken as an indication that the calibration process did not succeed. In the results of Setup B, it can additionally be seen that the RMS distance of the point cloud to the calibration plane was unrealistically small, being only around 0.5 mm. This value is even smaller than the standard deviation of the distances in one fixed direction (more than 0.8 mm in this case); this emphasizes that these results were not plausible.

We often observed during the optimization process that led to implausible results that the optimization algorithm tilts the two mirrored point clouds in such a way that the fitted calibration plane is parallel to the viewing direction of the Lidar. As a result, the present distance noise in the data did not result in an RMS distance of the point cloud to the calibration plane any more, since this distance noise now occurred along the extension of the plane. An unrealistically low value of the RMS distance to the plane is a good indicator of this happening. The same problem also occurred when the calibration procedure was executed without the mirror distances fixed or without the additional constraint of the two Lidar directions of the retroreflection foil to coincide. In general, these far-off calibration results were due to the calibration problem being numerically under-constrained.

The calibration setups C, H, and I all produced calibration results that we considered reasonable. In these cases, the numerical optimization converged to a realistic geometric scenario. Looking at the details of those successful calibration setups, those with maximal tilt of the calibration target (Setups C, F, and I) were the most successful (except for Setup F). At the largest distance, the setup with a target tilt around only one axis (Setup H) also succeeded. All successful calibration setups had in common that the Lidar points covered a large width on the calibration plane. This large width can be achieved by a large distance between the target and the Lidar or, alternatively, by tilting the calibration plane such that the Lidar points have a larger distance on one side of the calibration plane. Furthermore, a rotated calibration plane was observed to have an additional advantage (e.g., Setup G versus H) over calibration plane orientations that are symmetric to the yz-plane.

To evaluate the measurement error of the determined mirror angles, the mean value of those mirror angles was calculated, as well as the standard deviation, but only for the three successful calibration setups (C, H, I). This gave (mean ± standard deviation):
Left mirror: β_l_ = 23.74° ± 0.29°, δ_l_ = 33.62° ± 0.57°;Right mirror: β_r_ = −24.76° ± 0.18°, δ_r_ = 32.64° ± 0.55°.

This shows that the presented calibration method was capable of yielding reliable angles of both mirrors when taking care that the calibration target was positioned at a distance of ~600 mm from the Lidar sensor and was additionally tilted. In this case, the uncertainty of the mirror angles was around 0.5°. This angle uncertainty translated into a systematic transversal distance uncertainty of around 5 mm at an object distance of 600 mm (where the calibration target was used) and of 18 mm in the maximum measurement distance of 2 m of the current Lidar sensor. This is already smaller than the specified longitudinal systematic distance uncertainty given by the Lidar manufacturer of 40 mm and, therefore, considered good enough for the purpose of precise environmental perception in front of a robot vehicle. Additionally, this resulting angle uncertainty of only 0.5° was much smaller than the error of the angles taken directly from the CAD.

An example of the 3D point cloud data generated from such a calibrated mirror setup on a Lidar sensor is shown in Figure 8.

In order to make sure that the above calibration results not only produced precise measurement data when used on the same geometric object that was used during calibration, but also resulted in precise data when used on different objects, the following verification experiments were conducted: A first point cloud dataset was acquired and saved. This dataset was used to calibrate the mirror positions and orientations as described above. The resulting mirror angles and positions from this calibration were then taken as fixed during the following steps. Then, a second dataset was taken with the calibration plane positioned at a different position or in a different orientation. The calibration algorithm was used for a second time, but only the position and orientation of the calibration target were set as free for the optimizer during this second run. As a result, we could check how accurate the mirror calibrations values from the first (calibration) run could transform point cloud data from a different obstacle setup in the second (verification) run. To quantify the results of this verification experiments, the following Table 5 shows the resulting RMS deviations of the calibration plane distance after the verification run, as well as the best-fit orientation angles of the calibration plane after the verification run.

The presented verification data in Table 5 show that the RMS distances were only in the range of 1 mm. This showed that, during the verification run, the parameters still properly converged and no significant additional distance error was present. The presented plane angles in Table 5 fit well to the known orientations of the plane object in the verification dataset. The observed small deviations from those angle values were due to the theoretical values only being known to an uncertainty of around ±7°, as well as a different sign of the β_target_ angle, which was compensated for in the table. Therefore, the selected calibrations C, H, and I proved to be correct also when used on point cloud data not seen during the calibration process. As a result, we could show that the presented calibration method is suitable for the calibration of the mirror positions and orientations for use with a 2D-Lidar sensor.

## 4. Discussion and Conclusions

This research paper first described a prototype vehicle that used a 2D-Lidar setup with two mirrors to detect the position of the ground it drives on. It can use this information to avoid driving over obstacles or over edges. The design of this vehicle is not new in itself; similar designs and performance have already been presented earlier [8,9]; however, this prototype vehicle was the basis for the following new experiments.

This article explored a variety of different geometric design possibilities for reshaping the FOV of a 2D-Lidar with two plane mirrors. A few of these design variants were also mentioned earlier [8], but to our knowledge, this is the first systematic overview of different geometric setups in order to detect the ground under an AGV with a 2D-Lidar and plane mirrors. For several typical geometrical configurations, the resulting FOV shape on the ground was demonstrated. The geometric setup can be optimized for minimum mirror weight, minimum vehicle width, a short distance of the ground FOV in front of the vehicle, or a large width of the FOV in front of the vehicle. We suggest that the driving properties of the vehicle, such as driving speed and turning radius, are most important for determining which of those FOV shapes is most appropriate for the intended use. A closer FOV on the ground is advantageous for slower vehicles and a wider FOV on the ground for vehicles with a smaller turning radius.

As our main result, we presented a new method to calibrate the positions and angular orientations of the plane mirrors attached to a 2D-Lidar (Section 2.4). To the best knowledge of the authors, this is the first calibration method presented for this kind of mirror setup around 2D-Lidars. Earlier publications either did not deal with this problem (and, presumably, just used coarse CAD values for the mirror angles) or focused exclusively on the calibration of plane mirrors next to a 3D-Lidar sensor. Comparing our calibrated mirror angles values in Section 3.3 with the theoretical CAD values from Table 4, one can see a difference of around 2° between them. This underlines the need for such a calibration procedure in order to obtain precise point cloud data when using mirror setups on a Lidar sensor. A lack of calibration leads to systematic measurement errors of the point cloud data otherwise.

In this publication, we presented our systematic experiments to find the best parameters to make the presented calibration method succeed (Section 3.3). We found that for optimal calibration results, the calibration plane should have a large distance and ideally also be tilted around both axes. For the 2D-Lidar sensor used in this experiment, which has a 2 m measurement range, a distance of 600 mm from the calibration target was shown to be sufficient. We could show that with the presented prototype vehicle, the resulting accuracy of the point cloud data was improved after performing the calibration, compared to using the angular mirror positions from the CAD program directly without any calibration. It was additionally shown that the resulting point cloud data after the geometric transformation with these calibrated mirror orientations had a sufficient accuracy compared to the typical inherent distance noise of Lidar sensors. The distance error of the point cloud data did not significantly change after the calibration process. Finally, the presented calibration procedure yielded mirror pose results that also still worked fine on point cloud data not seen during the calibration process. We therefore concluded that the presented calibration procedure is a useful and robust process.

The presented calibration procedure is easy to perform in several aspects: The necessary steps during the calibration procedure are easy to conduct. Only materials that are easily available to both industrial manufacturing companies of AGVs, as well as to robot researchers are needed for the calibration procedure. Importantly, the shape of the main calibration object used is a simple plane. This allows for easy upsizing of this calibration procedure to 2D-Lidars with a longer measuring range by simply using larger calibration planes. Manufacturing any other shape of calibration object would probably pose much bigger problems in the necessary dimensions of several meters.

Additionally, the presented calibration procedure is general in the sense that it could easily be adopted for a different mobile robot using a different 2D-Lidar sensor. In case a different Lidar sensor should not have the possibility to read out intensity values in addition to the distance values, we outlined a variation of the calibration procedure in Section 2.4 that substitutes the reflector foil in our setup with a hole in the calibration plane object. We predict this procedure variant to succeed for Lidar sensors without intensity data output. The plastic material of our calibration object is not relevant: the calibration could be performed also with any plane object of a different material. Otherwise, our developed calibration methodology does not make use of any geometric or other peculiarity of our prototype vehicle or 2D-Lidar sensor. Therefore, we predict that this calibration method will succeed as well on mobile robots of a smaller or bigger size, as well as using different 2D-Lidar sensors or different calibration plane materials.

As a limitation of our experiments, the rather short measuring range of 2 m of the Lidar sensor should be mentioned. For using our calibration method on Lidar sensors with a larger measurement range, it would be necessary to increase the distance and the transverse dimensions of the plane measurement target. However, such distances and large transverse extensions could probably easily be realized with the proposed calibration setup, with a wall of a room or even an outside wall of a building taken as flat calibration target. We hypothesize the presented calibration method to be easily upscalable.

Further research using a 2D-Lidar sensors with a much larger measurement range, e.g., 15 m, is desirable, because it could enable the presented technique to be used in mobile robots that have a higher velocity than indoor AGVs, e.g., for outdoor settings where objects at further distances are relevant for robot navigation and autonomous vehicles. As a second area of future research, performance figures on the obstacle detection rate and overall functionality of a mobile robot in its real operation area with an obstacle avoidance algorithm using our ground detection and calibration method would be of interest. A third area of further research is the development of an open-source software library to assist robot developers by automating the calibration procedure described in Section 2.4 and by performing the real-time transformation of the point cloud data using the calibrated mirror parameters. Additionally, a very interesting topic for further research is to generate a fully analytical parametrization of the mirror geometry in this setup. Analyzing these analytical expressions could lead to a deeper insight into why the two additional constraints included in our calibration procedure are necessary: the mirror distance measured directly when covered with paper and the reflector foil attached at the center of the calibration target.

To draw a conclusion, we showed that a 2D-Lidar sensor with additional plane mirrors significantly improved the FOV of an ARV to achieve both environmental distance perception and ground detection without loss of data accuracy and without great expense. The manufacturing costs and engineering effort of such a setup are much smaller than for a 3D-Lidar sensor or other competing sensors, and still, this setup delivers the necessary point cloud data to enable robots to perceive the ground they are driving on or an obstacle they are flying close to precisely. The presented calibration procedure ensures that the geometric information perceived by the autonomous robot has a high precision. In the presented prototype vehicle, a measurement uncertainty of around 15 mm was achieved, which is comparable to the performance of current 3D-Lidar sensors. This improves ARVs’ capabilities to navigate and act in spaces that are not exclusively prepared for robot activity and where human activity can lead to unexpected obstacles such as objects on the floor, as well as unavoidable obstacles such as downward stairs. It hereby enables the development and production of more cooperative autonomous robots, for example for transport in hospitals, assistance in nursing homes, and a multitude of novel fields of application.

## Figures and Tables

**Figure 1 sensors-22-07830-f001:**
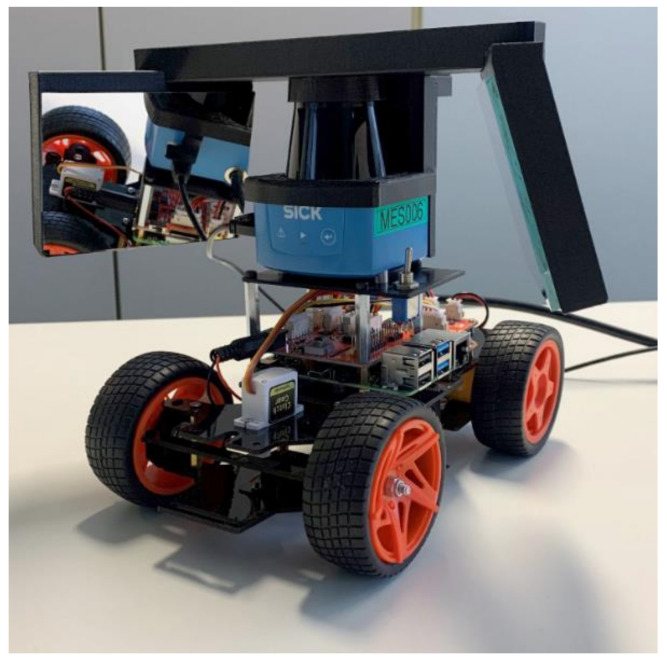
Prototype vehicle. As the construction base, the “PiCar” can be seen, consisting of four rubber tires, a servomotor for steering the two front wheels, and a “Raspberry Pi” computer for data processing and navigation. A mechanical platform holds the 2D-Lidar sensor (blue) on top of the computer board. Attached to the Lidar sensor is a larger mechanical part (black), which holds the two plane glass mirrors at either side of the Lidar sensor. Lithium polymer batteries and the power supply board are not visible as they are attached underneath the vehicle.

**Figure 2 sensors-22-07830-f002:**
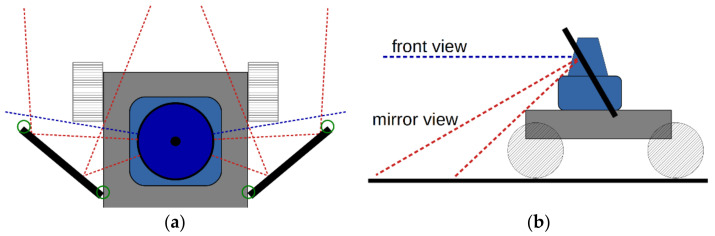
(**a**) Top view and (**b**) side view of the geometry for the geometric design simulations. In the center of the image, the 2D-Lidar sensors can be seen (blue). The black lines represent the two plane mirrors. The green circles at their ends have a diameter corresponding to the light receiving aperture of the Lidar sensor. The red lines represent the light beams, or observation directions, of the Lidar sensor. They all originate from the center point inside the Lidar.

**Figure 3 sensors-22-07830-f003:**
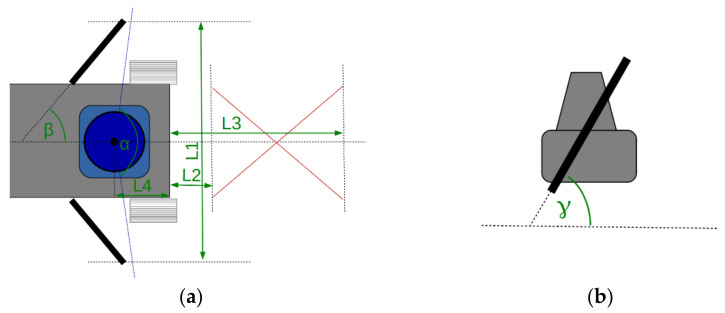
Sketch of the geometry parameters of the mirror configuration (green). The mirror FOV on the ground is sketched in red color. Top view (**a**) and side view (**b**). Note that the γ angle in the side view is defined with respect to the local “front“ direction of the mirror and not in the global coordinate system.

**Figure 4 sensors-22-07830-f004:**
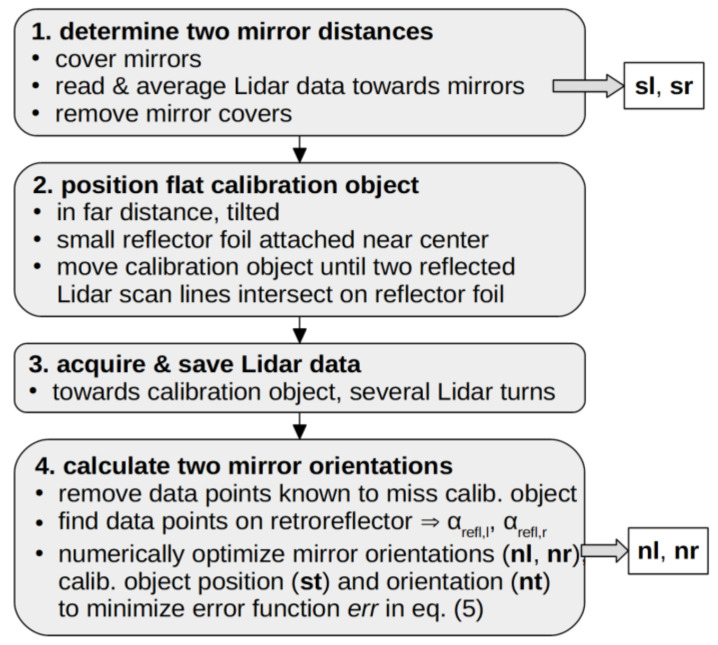
Flow chart of the calibration procedure. These four steps were conducted to calibrate the positions (**sl**, **sr**) and orientations (**nl**, **nr**) of the two plane mirrors next to the 2D-Lidar sensor. Details on each step are presented in Section 2.4.

**Figure 5 sensors-22-07830-f005:**
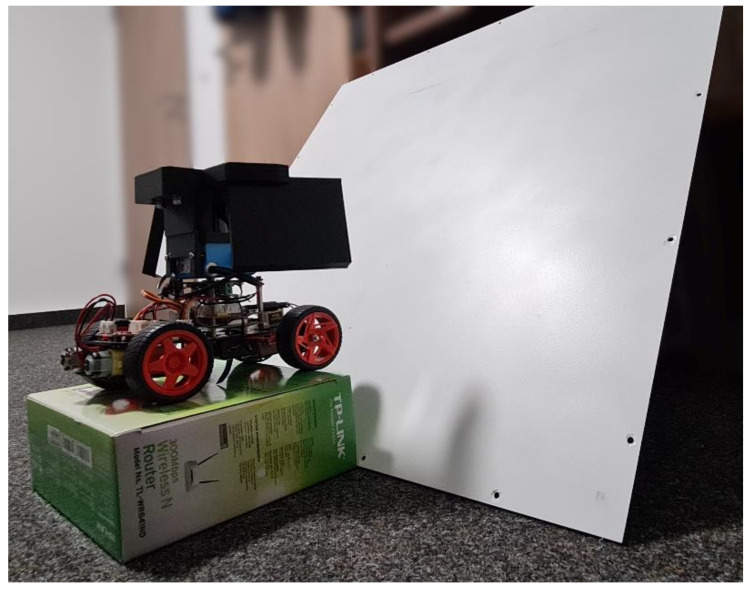
Photograph of the calibration setup. A white flat surface is positioned in front of the prototype vehicle. The position of the vehicle is elevated in order to have the Lidar sensor detect a larger part of the surface. The calibration surface is tilted around the x-axis for better calibration results. Note the 10 mm square piece of reflector foil that is attached near the center of the white surface and is necessary for the calibration procedure to succeed. (Background of room blurred for privacy.)

**Figure 6 sensors-22-07830-f006:**
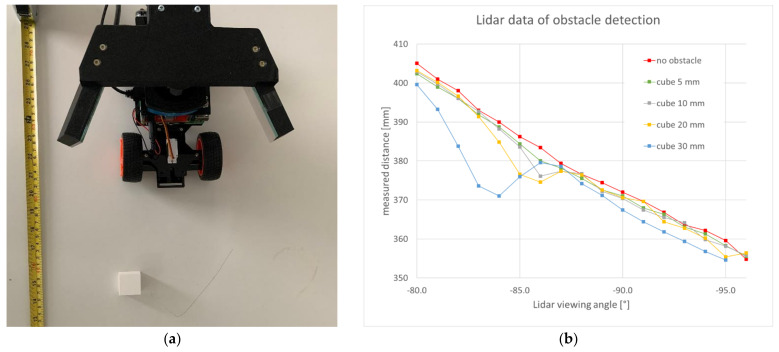
(**a**) Measurement setup with a cubical obstacle placed in front of the prototype vehicle. The Lidar setup with two mirrors acquires distance data in this setup to check what sizes of obstacles can be reliably detected. The presented setup is also a typical example for obstacles that severely threaten the prototype vehicle and keep it from driving on, if not detected and avoided. (**b**) Measurement data: Lidar distance raw values over Lidar rotation position for several sizes of cube obstacles in front of the prototype vehicle.

**Figure 7 sensors-22-07830-f007:**
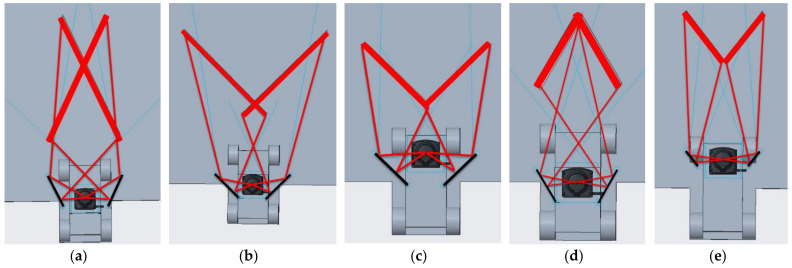
Top views of typical resulting CADs for the five design goals 1 to 5. The displayed geometries in (**a**–**e**) correspond to the listed geometry parameters 1 to 5 in Table 3. The thick red lines show where the Lidar laser scans the ground in front of the vehicle, and the thin red lines show the laser beam path from the Lidar center towards both ends of the thick red lines on the ground.

**Figure 8 sensors-22-07830-f008:**
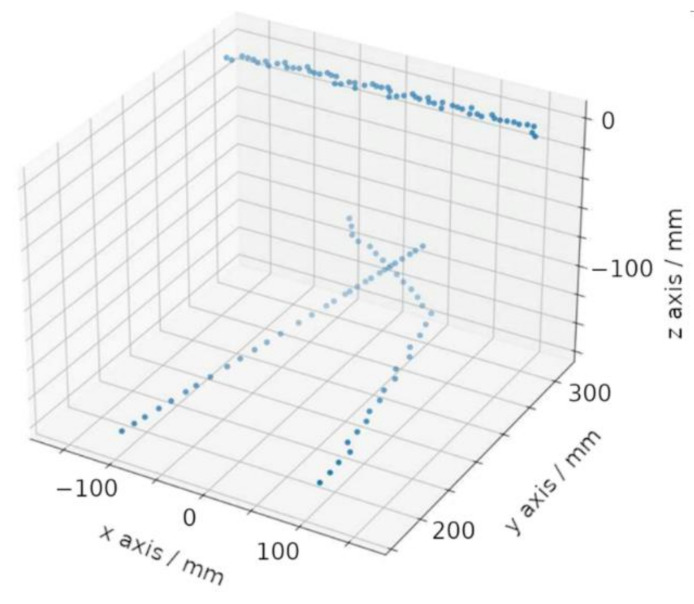
Example of resulting 3D point cloud data acquired by the Lidar sensor with two mirrors. The prototype vehicle had a flat wall in front for the data acquisition. Data points on this wall, as well as on the floor surface can be seen.

**Table 1 sensors-22-07830-t001:** Geometry parameters used in experiments with the prototype vehicle. Definitions of the geometry parameters are shown in Figure 3. Alternative geometry setups, as well as a visualization of the geometry setup are presented in Section 3.2.

Geometry Parameter	α	β	γ	L_1_	L_2_	L_3_	L_4_
CAD value used in prototype vehicle	110.2°	25.1°	73.5°	214.4 mm	81.1 mm	419.8 mm	100 mm

**Table 2 sensors-22-07830-t002:** Experimental results for the FOV position on the ground in front of the prototype vehicle. Definitions of the position values are in Figure 3; theoretical CAD values are identical to Table 1.

FOV Position Value on Ground in Front of Vehicle	Theoretical CAD Value	Measurement Result
min. distance L_2_, left side of vehicle	81.1 mm	53 mm
min. distance L_2_, right side of vehicle	81.1 mm	62 mm
max. distance L_3_, left side of vehicle	419.8 mm	368 mm
max. distance L_3_, right side of vehicle	419.8 mm	425 mm

**Table 3 sensors-22-07830-t003:** Resulting geometry parameters for typical CADs aiming for each of the five different design goals. (See Figure 3 for definitions of the used geometry parameters.).

Design Goal	α	β	γ	L_1_	L_2_	L_3_	L_4_
1	110.2°	25.1°	73.5°	214.4 mm	81.1 mm	419.8 mm	100 mm
2	145.4°	45.0°	73.0°	220.8 mm	96.7 mm	337.1 mm	100 mm
3	180.0°	46.2°	67.0°	223.3 mm	59.6 mm	237.0 mm	34.8 mm
4	135.6°	29.8°	73.0°	159.3 mm	85.0 mm	239.0 mm	100 mm
5	144.6°	38.0°	70.0°	167.9 mm	182.8 mm	290.4 mm	34.8 mm

**Table 4 sensors-22-07830-t004:** Calibration results of the different calibration setups. The resulting quality criteria, as well as the resulting orientation angles of the two mirrors are given. Two lines per calibration setup represent two independent measurements to illustrate the stability of the results. The first line shows results for a calibration performed on a dataset acquired for the stated geometric setup and the results calculated from this dataset. The second line shows these results with the calibration and evaluation procedure repeated on a second independent dataset for the very same geometric setup.

Calibration Setup	RMS Distance	Mean Intensity(RSSI)	Distance st. dev. Front	Right Mirror		Left Mirror	
				β_r_	δ_r_	β_l_	δ_l_
CAD (theory)	-	-	-	−25.07°	34.96°	25.07°	34.96°
A	1.05 mm	142.46	1.25 mm	−26.73°.	15.25°	24.99°	15.82°
	1.22 mm	142.29	1.19 mm	−25.34°	25.29°	23.76°	26.18°
B	0.55 mm	143.54	0.97 mm	−26.83°	9.82°	24.97°	10.25°
	0.40 mm	140.50	0.79 mm	−26.95°	6.07°	25.12°	6.32°
C	1.18 mm	156.25	1.17 mm	−24.57°	33.37°	23.36°	34.48°
	1.30 mm	155.25	1.19 mm	−24.67°	32.39°	23.47°	33.49°
D	1.17 mm	159.73	0.98 mm	−25.37°	29.51°	23.83°	30.55°
	1.18 mm	158.44	0.89 mm	−25.07°	30.58°	23.66°	31.61°
E	1.07 mm	158.68	0.70 mm	−25.52°	28.86°	23.73°	30.00°
	1.03 mm	158.25	0.76 mm	−25.14°	31.19°	23.48°	32.33°
F	1.53 mm	175.41	0.74 mm	−23.45°	37.37°	23.63°	37.25°
	1.35 mm	174.06	0.83 mm	−23.61°	37.03°	23.56°	36.97°
G	1.65 mm	178.87	1.18 mm	−24.44°	28.91°	23.10°	30.08°
	1.05 mm	180.81	1.11 mm	−24.63°	22.29°	24.32°	22.84°
H	1.11 mm	168.35	1.06 mm	−24.60°	33.30°	24.01°	34.21°
	1.03 mm	168.86	1.06 mm	−24.79°	32.44°	24.10°	33.35°
I	1.23 mm	170.79	0.90 mm	−25.00°	32.18°	23.72°	33.22°
	1.30 mm	170.69	0.69 mm	−24.93°	32.14°	23.75°	33.16°

**Table 5 sensors-22-07830-t005:** Verification results for the three successful calibration setups (C, H, I). The column with the verification setup also shows the approximate real angles of the used plane object. The RMS distance variation of the point cloud data on the plane object is given after the verification run. Additionally, the resulting orientation angles of the plane object of the verification run are given. (The plane object used for the verification runs was the same as the calibration plane object used for the calibration runs.)

CalibrationSetup	Verification Setup	RMS Distance	Distancest. dev. Front	Plane Object Angle 90° + β_target_	Plane Object Angle δ_target_
C	D (~45°,~0°)	1.27 mm	0.98 mm	39.60°	−2.15°
H	I (~45°~,20°)	1.57 mm	0.69 mm	36.56°	21.67°
H	A (~45°,~0°)	1.55 mm	1.25 mm	39.42°	1.38°
I	H(~30°, ~0°)	1.10 mm	1.06 mm	25.97°	−0.60°

## Data Availability

Not applicable.

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
