# Peer review of "Design and Calibration of Plane Mirror Setups for Mobile Robots with a 2D-Lidar"

_sensors, 2022, doi:10.3390/s22207830_

Round 1
Reviewer 1 Report
The authors wrote an interesting work in the field of environmental perception on autonomous robotic vehicles. I would like this article to be useful not only to the authors themselves, but also to those specialists who work on this problem because the text is difficult for reading in some parts. It seems to me that taking into account the comments and recommendations given below will contribute to the achievement of this goal.
Comments and Recommendations:
1. All formulas in the article are numbered. But almost none of them are mentioned in the text. It's not clear why formulas should be numbered if they are not referenced in the article?
2. The spelling of the abbreviation errRMS must be the same in formula (5) and line 352.
3. Formulas (1) – (4) are not mentioned anywhere in the text. Apparently, they are given to clarify the geometry of the problem. But it is very difficult to understand it without an explanatory drawing, which would be very appropriate in this part of the manuscript.
4. The authors give vector designations (for example, "sl, dl", so on), which are then not used anywhere. The question is, for what purpose is this being done? To impart some science?
5. The authors cite photographs (1,4,5a) to enhance visibility. But what the reader should pay attention to, the authors do not report. The discussion of these drawings should be more detailed, it is necessary to indicate what conclusions should, in the opinion of the authors, be drawn from these illustrations.
This paper is well enough written to understand main results. The manuscript seems to be appropriate for publication. I am therefore inclined that such a work corresponds to the content of the Journal "Sensors" and can be published there after minor mentioned corrections.
Reviewer 2 Report
The paper presents in high detail the calibration procedure of 2d-Lidar sensors for mobile robots using plane mirrors.
The quality of the work is great, but there are a few aspects which might improve its quality.
Can you comment on the generality degree of the developed methodology? How easy would be for someone else, using a different mobile robot and maybe different materials to use this method?
Can you comment on the efficiency of the proposed method? In numbers, how would you rate the improved detection and overall functionality of the mobile robot (with and without this calibration method)? The authors mention only few previous papers which performed lidar sensors calibration. Why hasn't been this aspect thoroughly studied before?
A lot of numerical calculation is performed and presented. Little parametrization is given. Is it not necessary?
Small comments:
Can you make an overall information flow chart with regard to the proposed methodology? It would increase the readability of the paper.
Figure 1 - please indicate the major components
Same page: “The green circles at the ends of the mirrors …” Please reformulate, it does not sound very “engineering”
Same page: “For this reason, the CAD simulation takes care…” How does it? Please reformulate
Some other minor English phrasing should be checked.
Reviewer 3 Report
1- The abstract needs more interest and rewriting some paragraphs.
2- There are still some aspects that can be improved (for grammar and punctuations). Improve the technical writing of your paper, where there are several grammatical errors and spelling I think they need to be checked out.
3- The conclusion needs more efforts to elaborate the achieved results with respect to the future work,
4- The practical part is very important,
5- Future work is an important part of the conclusion.
Round 2
Reviewer 2 Report
The authors have addressed all the issues I have pointed out previously. I think the paper may be published
Reviewer 3 Report
The paper no is good